# Tips to Quit Smoking: Perspectives from Vietnamese Healthcare Providers, Community Leaders, and Past Tobacco Users in the United States

**DOI:** 10.3390/ijerph20126160

**Published:** 2023-06-17

**Authors:** Tina N. Le, Shweta Kulkarni, Michael S. Businelle, Darla E. Kendzor, Amanda Y. Kong, Anna Nguyen, Thanh Cong Bui

**Affiliations:** 1TSET Health Promotion Research Center, Stephenson Cancer Center, University of Oklahoma Health Sciences Center, Oklahoma City, OK 73104, USA; shweta-kulkarni@ouhsc.edu (S.K.); michael-businelle@ouhsc.edu (M.S.B.); darla-kendzor@ouhsc.edu (D.E.K.); amanda-kong@ouhsc.edu (A.Y.K.); thanh-c-bui@ouhsc.edu (T.C.B.); 2Department of Biostatistics and Epidemiology, Hudson College of Public Health, University of Oklahoma Health Sciences Center, Oklahoma City, OK 73104, USA; 3Department of Family and Preventive Medicine, College of Medicine, University of Oklahoma Health Sciences Center, Oklahoma City, OK 73117, USA; 4Fran and Earl Ziegler College of Nursing, University of Oklahoma Health Sciences Center, Oklahoma City, OK 73117, USA; anna-nguyen@ouhsc.edu

**Keywords:** smoking cessation, tobacco treatment, cancer prevention, Vietnamese population

## Abstract

This study focuses on smoking-cessation strategies for United States (US) Vietnamese individuals, a group with high smoking rates, particularly those with limited English proficiency (LEP). The researchers conducted 16 in-depth interviews with a diverse group of participants, including healthcare professionals, community leaders, and former tobacco users. Data were analyzed using the Phase-Based Model of smoking cessation, resulting in several helpful strategies across the four phases: Motivation, Preparation, Cessation, and Maintenance. Prominent advice for the Motivation Phase included having a strong determination to quit and a reason why, such as protecting loved ones. For the Preparation and Cessation Phases, participants recommended healthy coping mechanisms, avoiding triggers, changing habits, and gradually reducing the number of cigarettes smoked. In the Maintenance Phase, strategies included regular exercise and setting boundaries with other people who smoke. Participants also stressed the importance of social support throughout all four phases. These findings have implications for healthcare providers working with US Vietnamese who smoke, especially those with LEP. By understanding the unique challenges this group faces in accessing smoking-cessation resources, providers can offer tailored support and guidance. Ultimately, this study provides useful strategies for helping US Vietnamese quit smoking, improving their health outcomes and quality of life.

## 1. Introduction

Tobacco use is estimated to cause over 480,000 deaths annually and is the single largest preventable cause of death in the United States (US) [1]. Tobacco use causes multiple negative health consequences, including cardiovascular disease, respiratory disease, and cancer [1]. Thus, smoking cessation is an evidence-based strategy to decrease tobacco-related morbidity and mortality rates [2,3,4].

One systematic review showed that lung-cancer patients who are US Asian immigrants have worse 2-year survival than those Asian individuals born in the US [5]. Among the Asian population in the US, Vietnamese men have the highest smoking prevalence (29.5%) compared to Chinese (23.6%), Filipino (24.4%), Other Asian (18%), and Biracial/Mixed (23%) [5,6,7]. Smoking prevalence within the US Vietnamese population is higher among Vietnamese men with limited English proficiency (LEP) (32.7%) compared to those with higher English proficiency (25.7%) [7,8,9]. Among Asian and Pacific Islander populations in the US, Vietnamese immigrants have lower educational attainment and worse health status [10], and compared to other Asians and Pacific Islanders, lung cancer [5,11] and bronchial cancer rates [5] are highest for Vietnamese people. These observations all underscore that it is critical to support US Vietnamese people in quitting smoking to prevent tobacco-related morbidity and mortality.

Resources in the Vietnamese language to help US Vietnamese people who smoke to quit (e.g., the Asian Quitline—https://www.asiansmokersquitline.org/, accessed on 27 May 2023) are very limited. Efforts to adapt and disseminate theory-based smoking-cessation interventions and to utilize mobile health (mHealth) interventions for smoking-related cancer prevention in the US Vietnamese population are lacking. To address this need, our parent pilot study aimed to utilize mHealth technology to promote smoking cessation and cancer screening for US Vietnamese cigarette users with LEP. To develop mHealth-based intervention content that is culturally relevant to the US Vietnamese, we conducted a qualitative study with US Vietnamese healthcare professionals, community leaders, and past tobacco users to explore applicable strategies and techniques, as well as to identify tips to help US Vietnamese cigarette users quit smoking. We use the term “US Vietnamese” (instead of “Vietnamese Americans”) because our study aims to target a broader population of Vietnamese people in the US who may not be US citizens (e.g., permanent or non-permanent residents, or those who are in the US on working or non-immigrant visas).

## 2. Materials and Methods

This report follows the Standards for Reporting Qualitative Research (SRQR) [12]. We conducted 16 in-depth interviews (IDIs) with six US Vietnamese healthcare providers, five community leaders, three past tobacco users, and two participants who were both healthcare providers and past tobacco users. We recruited participants through our network and connections with the Vietnamese organizations and clinics that serve the Vietnamese community across four US states: Oklahoma, Connecticut, North Carolina, and Florida. The inclusion criteria included that participants must be at least 18 years of age and be able to provide written informed consent. The average length of IDIs was 60 min, and all IDIs were conducted via Zoom^®^ (Zoom Video Communications, Inc., San Jose, CA, USA) in either English or Vietnamese, based on the participant’s preference. Interviews were recorded and transcribed verbatim. The study was approved by an Institutional Review Board of the University of Oklahoma Health Sciences Center (#12713).

The IDI questions for healthcare providers focused on when and how they helped others to quit smoking, what strategies were effective, and challenges in the process of quitting smoking. The IDI questions for past tobacco users included their tobacco-use history and their smoking-cessation journey, including what specifically helped them quit and stay abstinent. Since they have broad experience with Vietnamese culture, often interact with Vietnamese people, and understand their social behaviors, we also interviewed Vietnamese community leaders to gain their perspectives on smoking cessation.

Research-team members analyzed data in both languages, with the aid of MAXQDA software, using English codes and themes. There were three bilingual coders with a graduate background in health sciences (pharmacy, nursing, and public health). The main coder, who is fluent in both English and Vietnamese, analyzed both the original English and Vietnamese transcripts; one secondary coder reviewed the coded transcripts in English; and one secondary coder who is more fluent in Vietnamese reviewed the coded transcripts in Vietnamese. Discrepancies were discussed among coders until a consensus was reached. A thematic content analysis approach was used. Codes and themes were informed by constructs and phases of the Phase-Based Model (PBM), a behavioral theory specific to smoking cessation [13,14]. PBM includes four phases: Motivation (readiness to quit), Preparation (planning to make a quit attempt), Cessation (making a quit attempt and coping with challenges such as withdrawal symptoms), and Maintenance (continuing to stay abstinent); these serve as themes for data analysis. [13,14]. We chose PBM as a guiding theory for analyzing and identifying strategies that address different challenges between each phase of smoking cessation.

## 3. Results

During interviews, participants identified their age, sex, profession, and specialty. The healthcare providers’ professions and specialties were geriatric pharmacist, social worker, retail pharmacist, dental intern, nephrology nurse practitioner, clinical pharmacy resident, pharmacy faculty, and nursing faculty. Of the sixteen total participants, nine were men (~56%) and seven women (~44%). Overall, 75% of healthcare providers were women, and 80% of community leaders were men. The age range of healthcare providers was 28-to-49 years, community leaders were 46-to-71 years old, and past tobacco users were 39-to-69 years old.

### 3.1. Motivation Phase

The most salient theme for the Motivation Phase is the need to have a firm determination to quit or a reason why smoking cessation is necessary. Some common motives or reasons are described below.

#### 3.1.1. Protecting the Family or Children from the Harms of Smoke

Most past tobacco users reported worrying about their children’s health as a motive to quit. One participant stated, “*What motivated me was when I had my first son and he had borderline asthma… Although I went outside to smoke, the smoke was still on my clothes, so when I came in I could not hold him anymore… I [might inadvertently] put him in jeopardy [by smoking]… Having a newborn son had motivated me the most.” He continued, “…Before, I knew cigarettes were bad but I never wanted to quit because I thought smoking was fun but until then, I had a real motivation like my son, I was able to quit cold turkey, I didn’t need anything…*” He was determined to quit to protect his newborn son from secondhand smoke (male, clinical pharmacy resident and past tobacco user, 39 years old). Worrying about one’s health and family’s health is another prominent motive for quitting. Another past cigarette user decided to quit before getting married, stating that once he started a family, he wanted his household to be smoke-free. “*Quitting smoking is good for my own health… and the important thing is that I don’t want my future wife and children to have nicotine in their bodies. That was a strong goal for me to plan ahead and quit smoking*” (male, past tobacco user, 53 years old).

#### 3.1.2. Serving as a Role Model for Others

Healthcare providers are more prone to taking responsibility to follow the advice that they give patients and to serve as a professional role model. A 31-year-old male dental intern who formerly smoked stated, “*[In] dental school, I really wanted to quit because I started to learn about the harmful effects of tobacco to my lung, and especially to my teeth. We always told patients not to smoke. But I smoked myself, so I did not think that was the right thing to do. So, I tried to quit myself.*” The desire to lead by example and thus earn respect from others was also identified as a motivation to quit smoking: “*Working in a health program but if I myself smoke, who would listen to my teaching?*” (male, past tobacco user, 69 years old). Serving as a role model was identified as important both in the healthcare setting and within the family unit. A social worker recalled a story of her friend who quit smoking: “*He had a 21-year-old son living with him… His son started smoking at 18 after seeing the dad smoked… and his son took the dad’s cigarettes to smoke… The dad knew that smoking was not good for his son, so he wanted to quit to be a role model for his son to quit as well*” (female, social worker, 42 years old).

#### 3.1.3. Avoiding Negative Health Consequences of Smoking

Many participants mentioned that they quit smoking themselves or advised others to quit because they wanted to prevent negative health consequences. One past tobacco user recalled a time when his friend told him, “*My dad is an oncology doctor, he said that the process of dying due to lung cancer is extremely painful. If you die because of other diseases, the pain of dying process might be milder, but the pain due to cancer, especially lung cancer, the patient would have to suffer through the agony of pain for a while before dying.*” Based on this conversation, he finally decided to quit smoking because he would rather “*die in peace*” than “*die in agony*” (male, past tobacco user, 69 years old). The motivation to quit smoking was described as greater when an individual’s own health was at risk or compromised, with one male community leader stating: “*When they feel like they are getting sick and go to the doctor, that’s when the Vietnamese smokers start to think about quitting, or they would quit when they are already sick*” (male, community leader, 45 years old). Another reason that motivated a family member of one participant to quit smoking was the desire to have a longer life shared with loved ones: “*My dad quitted smoking because he wanted to watch his grandchildren growing up*” (female, geriatric pharmacist, 42 years old).

Healthcare professionals thought that an opportune time to advise patients to quit smoking was during a consultation about medical issues. Statements emphasizing this view included the following:

“*A pragmatic test result [about medical issues] is not a happy sign. So, the consequence of having a stroke and a heart attack becomes a fear factor. Then the smoking cessation resources become more important to them*”.(female, nursing faculty, 49 years old)

“*Address all of their medical conditions, not just smoking cessation. We can explain to them about how smoking can worsen their lung, kidney, and heart conditions. For example, worsening heart failure could be a motivation to quit*”.(female, pharmacy faculty, 49 years old)

“*Smoking can cause periodontitis or gum disease which will eat away the bone. That is one of the reasons why people lose their teeth*”.(male, dental intern and past tobacco user, 31 years old)

#### 3.1.4. Spiritual/Religious Beliefs as a Motivation

Active spirituality was also reported as having helped former smokers to quit smoking. For example, one individual stated that “*Reading the Bible every day… all of a sudden, there was a conviction inside me that said I need to quit… God said that my body is my temple, and I need to keep it clean. So, I quitted 100% cold turkey. I did not have any problem ever since*” (male, dental intern and past tobacco user, 31 years old).

### 3.2. Preparation and Cessation Phases

Salient factors for the Preparation and Cessation Phases included healthy coping methods for stress, avoiding both smoking triggers and peer pressure, changing unmindful habits, and engaging in activities that distract from the craving for a cigarette. We grouped these two phases together since the identified techniques are applicable for both planning and implementing smoking cessation.

#### 3.2.1. Healthy Coping Methods for Stress

Stress was the most commonly identified reason for smoking across all groups. Ways to overcome acute stress included “going for a walk” or practicing meditation, thus avoiding smoking. One interviewee stated, “*When we are stressed, the body produces harmful hormones. So, when we keep the stress level down [by meditating], it’s healthier for the body*” (female, community leader, 71 years old).

#### 3.2.2. Physically Avoiding Smoking Triggers

Staying away from activities that participants associated with cigarette smoking, such as drinking alcohol, was mentioned as a necessary step to quit smoking. Participants also suggested that actively spending time in smoke-free environments and focusing on positive things in life could help eliminate both stress and the urge to smoke. Comments included the following:

“*…Especially when hanging out with each other, the Vietnamese guys would drink alcohol, and that makes them crave a cigarette, then they would smoke a lot, not just one cigarette. That’s a gateway for people to become addicted [to cigarette smoking], and is also a barrier for people to quit*”.(male, community leader, 49 years old)

“*I just got myself into a completely different environment… [go somewhere else] during the break at school. Then I went out of town for a month. After a month, I did not have any craving for smoking anymore... I physically tried to stay away from [smoking triggers] and stressful situations, go do something else [instead of smoking]*”.(male, dental intern and past tobacco user, 31 years old)

#### 3.2.3. Avoiding Peer Pressure

Peer-pressure in a social setting was a salient reported factor for why people smoke. The perceived polite gesture of smoking together or offering a cigarette to friends during social gatherings was seemingly a cultural norm among Vietnamese cigarette users, and it was identified as a challenge to not smoke when in social situations with other cigarette users. Specific comments included the following:

“*I wasn’t a regular smoker, but I used to smoke socially. When I met my friends and they offered me a cigarette, I just accepted and smoked with them… I wasn’t addicted to it and didn’t have to buy the cigarette. But you know, the difference between Vietnamese culture vs. the American culture about smoking is that smoking is considered a social grace or etiquette among Vietnamese people… When the Vietnamese meet each other, they often offer or share cigarettes for others to use freely. They would even invite others to smoke with them... If they only meet every one or two month, it’s fine. However, if this circle of people hangout every day and smoke like that, they will be addicted to cigarettes… In the Vietnamese culture, if you have cigarettes but don’t offer them to your friends, it’s considered selfish… That belief is the obstacle for the Vietnamese smokers to quit. The best way to avoid smoking [in this context] is to hang out with friends who don’t smoke*”.(male, community leader, 49 years old)

#### 3.2.4. Recognizing and Interrupting Cues to Smoke

Participants also discussed how certain objects or locations might cue their smoking behavior. A female pharmacist said she would advise a cigarette user that “*during breaks at work, you should avoid that specific location [that you usually smoke] and go somewhere else instead to distract the brain and prevent the autopilot urge to smoke.*” Another example that she provided was as follows: “*On the way home from work, one may smoke when they stop at a specific stop sign as a habit for many years. We can break that automatic habit by advising them to drive a different road so the brain would be distracted from that old habit [and will focus on the new road instead]*” (female, geriatric pharmacist, 42 years old).

#### 3.2.5. Engaging in New Activities 

Engaging in activities that are not associated with smoking was also discussed as a strategy to distract from the urge to smoke. Specific examples identified by participants included playing chess, going on a fun non-smoking trip, or chewing a candy or gum when craving a cigarette: “*Instead of finding a cigarette to smoke, one can find a candy, [eat] crackers or chew gums or something else to distract the mind from thinking about smoking*” (male, past tobacco user, 53 years old).

#### 3.2.6. Gradually Reducing the Number of Cigarettes Smoked 

With the ultimate goal of smoking abstinence, setting small reduction goals was suggested as an option for those who habitually smoke heavily (e.g., one pack per day or more) and are unable to quit immediately. One participant stated, “*If one smokes 1 pack per day, they can reduce 5 cigarettes daily [as the first goal]. Give them 3 or 4 weeks or a month depending on their preference. When they have achieved that goal, suggest to reduce another 5 cigarettes*” (male, past tobacco user, 53 years old).

### 3.3. Maintenance Phase

In the Maintenance Phase, relapse can be prevented by continuing a smoke-free, positive lifestyle. Participants identified several components of this lifestyle, including exercising regularly, healthy eating, being prepared to cope with the potential peer pressure at social gatherings (e.g., setting respectful boundaries around other people who smoke), considering quitting to be an important promise to keep (e.g., to children), and continuing to physically avoid triggers that remind one of smoking.

Participants believed that when someone has a strong reason to quit from the beginning, that person is usually able to maintain abstinence. One participant stated, “*[The tobacco users] need a true motivation for them to quit and then use that as an ongoing motivation…*” (male, pharmacist and past tobacco user, 39 years old). This would serve as both a motivation to quit and a driver to maintain the smoke-free state. Another participant identified avoiding social situations that trigger smoking, staying firm about the smoking-cessation decision/journey, and respectfully saying no to smoking invites as ways to prevent relapse: “*Avoid hanging around when other individuals are smoking. If I see my friends going outside to smoke, I don’t go outside with them. If I go with them, I will smoke for sure*” (male, past tobacco user, 57 years old). A past cigarette user identified hiding objects that remind him about smoking as another strategy to avoid triggers: “*…After [someone] stopped smoking, the lighter must be hidden [by a family member]. One can’t smoke if there is no lighter available*” (male, past tobacco user, 69 years old).

### 3.4. Factors Related to Multiple Phases of the PBM

Some salient factors were related or applicable to multiple phases. These included having a strong reason to quit, avoiding triggers or peer pressure, and adopting other healthy behaviors. Additionally, a prominent factor that was related to all four phases was creating an environment with repeated/continuous support from loved ones. A social worker (female, 42 years old) recalled the process of encouraging her friend to quit smoking: “*When I was working at the hospital… I told him that there were many sick people. Some people smoked and got this disease, others drank too much alcohol and got that disease… I told him [about the seriousness of different diseases due to smoking/drinking] everyday, so I think those stories started to permeate in his mind.*” When her friend relapsed, she texted him: “*It’s OK, relapse can happen. Don’t give up!*” She showed grace and empathy and stated that her friend felt supported and accepted. When her friend stayed abstinent for a week straight, she complimented him: “*You’re amazing!*” She believed that her words gave him even more motivation to continue staying abstinent. Repeated pressure from family members to quit smoking may affect someone’s thoughts and subsequent quitting behaviors. One past tobacco user described the following: “*My mom, my wife, my daughter kept telling me to quit… Whenever I hold a cigarette and hear [them nagging], the cigarette didn’t taste good anymore*” (male, past tobacco user, 57 years old). Another participant emphasized the effectiveness of repeated advice/encouragement: “*If my friend were to say that smoking would cause pain when dying once and then left, I would not have been scared [about the consequences of smoking]. My friend told me that almost every time she saw me smoke, so then I quitted completely*” (male, past tobacco user, 69 years old).

## 4. Discussion

In this qualitative analysis, we identified several applicable strategies and tips to help US Vietnamese individuals to quit smoking. The results of this qualitative study could provide important insights for developing culturally appropriate future interventions. The main strategy for the Motivation Phase was having a firm determination to quit or a reason for why smoking cessation is necessary. Many participants reported the need to protect their family from the harms of secondhand smoking and expressed the desire to avoid the negative health consequences as their motivation, with the ultimate goal being to live a healthy, happy life and spend more quality time with their family. This theme was consistent with previous studies finding “self-determination” or “willpower” as the most essential factors for an Asian individual to quit smoking [15,16,17,18]. Our study also indicated that awareness of how smoking cessation can impact other individuals, such as the younger generation or loved ones, can make one’s motivation stronger. We also found that motives for quitting smoking were highly individualized and may depend on an individual’s situation and experiences. Further research should explore how a healthcare provider or caregiver may be able to help Vietnamese individuals who smoke identify salient motives for quitting smoking, such as focusing his/her effort on further investigating the causes of smokers’ attachment to smoking.

Stress was one of the most mentioned reasons for individuals to smoke; smoking was identified as a coping method relevant to both the Cessation and Maintenance Phases. Other studies found that smoking ultimately increased everyday stress because most people who smoke would experience mild withdrawal symptoms between cigarettes, which built up psychological stress over time [19,20]. The risk regarding this negative feedback loop affecting the psychological state was greater than the temporary relief of stress from smoking. Thus, it is important for individuals who smoke to understand the long-term effect of smoking on stress levels and actively seek a healthier, non-smoking options while coping with stress during the Preparation and Cessation Phases. Study participants suggested strategies to reduce acute stress, such as meditation or taking a walk. Previous studies support that meditation and regular physical exercise are beneficial to individual well-being and help reduce stress [21,22]. Supporting this, mindfulness meditation can reduce blood pressure in people with hypertension [22]. To maintain abstinence, participants believed that it was important to keep a healthy, positive lifestyle throughout the smoking-cessation journey, consistent with a previous study indicating that a positive mindset can reduce the likeliness of relapse [23]. Thus, managing stress and cultivating a lifestyle with healthy habits could reduce the need for a cigarette as a stress-coping method.

Besides stress management, participants reported actively engaging in other activities, for example, playing chess or chewing candy or gum, to distract them from thinking about smoking. Additionally, participants suggested spending more time with friends who do not smoke and avoiding situations where people smoke. Participants mentioned Vietnamese people’s tendency to smoke together during social interactions, and they viewed this as peer pressure to smoke. They thought that Vietnamese people who smoke socially usually consider smoking together as a bonding gesture and a normal occurrence in social situations. Participants expressed that by understanding this social behavior related to Vietnamese culture, they could then remove themselves from the environment where they foresee that their friends may smoke. In a previous study, when a group of young adults who smoked were placed in an environment with people who did not smoke, the young adults ended up smoking fewer cigarettes than those who were not in the presence of people who did not smoke [24]. This indicates that people may be more likely to mirror the actions of peers whom they spend time with and represents a strategy to avoid smoking triggers. This shows that the surrounding environment plays an important role in impacting an individual’s effort to quit smoking.

A prominent factor that was related to all four phases of the PBM is creating a continuously supportive environment. Stories from our participants demonstrated that repeated encouragement helped individuals who smoke to quit. Previous studies have confirmed that having support from healthcare providers [25,26], partners [27], family [28], and friends [28,29] increases successful smoking cessation.

Pharmacologic intervention methods were rarely mentioned as a support for successful quitting by these participants. Only one healthcare provider mentioned nicotine patches as a tool to help a family member to quit. It was unclear whether that family member quit primarily because of the constant support from family, because of the medications, or because of a combination of both. A community leader mentioned that nicotine gum was ineffective for his father because he did not follow the recommended regimen. There is a need for more concrete direction and explanation from healthcare providers when counseling their patients on pharmacological interventions. Additionally, none of the past individuals who smoked reported using pharmacologic treatment during their smoking cessation. While self-determination alone has worked for many Vietnamese people who smoke [15,16,17,18], evidence-based pharmacologic interventions such as Chantix or nicotine replacement therapy also showed benefits in smoking cessation [30,31]. Medications, together with other strategies suggested by participants in this study, could help support Vietnamese individuals’ cessation outcomes.

This study has both strengths and limitations. A key strength is that this study included three groups of participants with different perspectives—healthcare providers, community leaders, and individuals who smoked in the past. Healthcare providers contributed the knowledge and counseling tips from their medical specialty, and those who were both healthcare providers and past tobacco users provided insights regarding the application of their medical knowledge in helping themselves and those around them quit smoking. Community leaders spoke about several factors related to Vietnamese culture, including peer pressure for smoking, while individuals who smoked in the past provided insights on experiences in their smoking-cessation progress. These perspectives offer a diversity of viewpoints and suggested strategies/tips during the smoking-cessation process. The limitations of this study include the different numbers of participants in each subgroup and that all of the people who smoked in the past in our sample identified as male. Future studies could explore the significance of such limitations. Despite these limitations, we did not observe any new themes upon interviewing each subgroup in this study.

## 5. Conclusions

This qualitative study identified several factors that serve as useful strategies to help US Vietnamese individuals quit smoking. The main strategy for the Motivation Phase was having a firm determination to quit or a reason why smoking cessation is necessary. We found that stress was one of the most mentioned reasons that cue individuals to smoke, which was relevant to both Cessation and Maintenance Phases. Participant strategies to quit included stress management and removing cues/avoiding triggers that remind one to smoke. Additionally, actively engaging in other healthy activities (e.g., exercising regularly and healthy eating) were suggested as strategies to prevent relapse. Another prominent factor that was related to all four phases of the PBM was creating and maintaining a supportive environment to quit smoking. Future research is needed that focuses on how healthcare providers can help Vietnamese individuals who smoke identify motivational factors and strategies to quit smoking and stay smoke-free.

## Data Availability

Data sharing not applicable.

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
