# Peer review of "Tips to Quit Smoking: Perspectives from Vietnamese Healthcare Providers, Community Leaders, and Past Tobacco Users in the United States"

_ijerph, 2023, doi:10.3390/ijerph20126160_

Round 1
Reviewer 1 Report
Overall:
-Consider the use of person-first language throughout – so “people who smoke” instead of “smokers” and “people who use tobacco” instead of “users”. See Tobacco Control’s new policy on this and accompanying editorial. https://tobaccocontrol.bmj.com/content/32/2/133.long
Title:
-This title runs the risk of being confused with a prominent national tobacco education campaign - consider revising
Introduction:
-line 38: Caution with this statement on health consequences – the listed health effects are related to cigarette smoking and, in the case of CVD and cancer, smokeless tobacco products, not necessarily nicotine itself. It’s important to thread the needle here so as to not mislead patients or providers about the health consequences of nicotine (and potentially dissuade them from use of NRT).
-Would be worthwhile mentioning the existence of the ASQ, which provides quitline services in Vietnamese (https://www.asiansmokersquitline.org/)
Methods:
-would be helpful to better understand how the participants were recruited
-would be helpful to better understand more specifically what the Vietnamese community leaders were interviewed about (beyond “perspectives on smoking cessation”)
-may be worthwhile explaining why you elected not to talk with people who currently smoke
Results:
-may be worthwhile explaining why there were no physicians or primary care providers included in the group of healthcare providers
-line 98- the age range of “past smokers” may have a typo as line 113 indicates the respondent from the quote was 39, and line 121 indicates a 31 year old who formerly smoked
-throughout the results, sometimes the respondent age is listed and sometimes it is not – recommend consistency in how this is presented
Discussion:
-line 301 – what are “light” abstinence symptoms?
-paragraph starting line 315 – many of the strategies noted (avoidance of situations where people smoke and other trigger situations, distraction techniques, etc.) are common strategies in cessation counseling – consider noting that this study reinforces that these techniques may also be applicable when support Vietnamese people who are trying to quit smoking
Reviewer 2 Report
I have read the article entitled " How to quit smoking: Tips from United States Vietnamese healthcare providers, community leaders, and past tobacco users". This is a very interesting study. It is worth, however, that the authors extend the introduction section to illustrate the problem a little better.
Author Response
The authors would like to thank you for your constructive comments and thoughtful critiques, which have helped to strengthen our manuscript. We have revised our manuscript according to the comments and suggestions. Changes or edited text in the manuscript are also highlighted using track changes in a supplemental copy, included in addition to the requested clean version of the manuscript.
Reviewer 2: I have read the article entitled " How to quit smoking: Tips from United States Vietnamese healthcare providers, community leaders, and past tobacco users". This is a very interesting study. It is worth, however, that the authors extend the introduction section to illustrate the problem a little better.
Response: We have extended the Introduction section with more details and clarifications. Thank you!
Reviewer 3 Report
This study is based on interviews, not a large sample size. The title therefore is a bit misleading as this is opinion based, not data or research based suggestions. The tips and insights are not novel, and this article doesn't really add much to the smoking cessation literature.
Nevertheless, descriptions are covered well and described adequately. I recommend for publication.
Author Response
The authors would like to thank you for your constructive comments and thoughtful critiques, which have helped to strengthen our manuscript. We have revised our manuscript according to the comments and suggestions. Changes or edited text in the manuscript are also highlighted using track changes in a supplemental copy, included in addition to the requested clean version of the manuscript.
REVIEWER 3: This study is based on interviews, not a large sample size. The title therefore is a bit misleading as this is opinion based, not data or research based suggestions. The tips and insights are not novel, and this article doesn't really add much to the smoking cessation literature.
Nevertheless, descriptions are covered well and described adequately. I recommend for publication.
Response: Thank you so much! To clarify the title, we have revised the title to “Tips to quit smoking: Perspectives from Vietnamese healthcare providers, community leaders, and past tobacco users in the United States.”
Reviewer 4 Report
This is an interesting and well-written article. The most compelling thing about the article is that it provides clear wisdom on how to actually quit smoking. The direct quotes from the respondents are helpful and make for engaging reading.
The Introduction is a bit weak. This should include information about other racial/ethnic groups and their efforts to quit smoking. As the article is currently written, the term "Vietnamese" could be substituted with the name of any other group. Are other groups just as concerned about health effects and effects on family members? How do efforts to quit smoking among Vietnamese differ from these other groups? Or are there no differences? Or do we not know about similarities and differences?
The Materials and Methods section is good. Sixteen respondents is an adequate sample. The explanation of how the coding was completed is good.
Another area of the article that needs strengthening is the very brief Section 3.1 Motivation phase. Here is a good opportunity to define motivation and its importance. There is so much literature on this and therefore there needs to be more discussion of how one becomes motivated to quit. How can family members and health care professionals help to motivate smokers to quit?
The subheadings under section 3.1 Motivation phase give a good understanding of the experiences of smokers: Protecting the family or children from the harms of smoke; serving as a role model for others; Avoiding negative health consequences of smoking; and Spiritual/religious beliefs as a motivation. The last motivation is really just one quote. Is there anything here that can be added from the literature that address spiritual/religious beliefs as motivators?
The discussion in the Preparation and Cessation phases is quite good. The Discussion section is also very strong. The bit of information about Vietnamese culture in the Strengths and Limitations section that pinpoints peer pressure is very important. I'd like to see more information about Vietnamese culture, and its bearing on smoking, infused throughout the manuscript.
The conclusion is strong--it reaffirms that relapse prevention is based upon cultivating a healthy lifestyle.
Overall, a good contribution to the literature!
Reviewer 5 Report
The topic of this article is imperative given the paucity of smoking cessation programs specific Asian ethnic groups such as Vietnamese Americans. The findings have the potential to help further guide how to create smoking cessation programs for Vietnamese Americans. Following are some comments to consider to further improve and strengthen the paper:
Introduction
-Along with the title, is there a reason why the target group is being referred to as United States Vietnamese as opposed to Vietnamese Americans? If there is a reasoning behind this, it should be made clear in the introduction.
-Throughout the manuscript, individuals who smoke were constantly referred to as “smokers.” I encourage the authors to consider using person first language to avoid stigmatizing people who use substances. A helpful resource on person-first language https://www.safeproject.us/resource/person-first-language/
-In the introduction, the authors discussed that Vietnamese Americans and Vietnamese American men have the highest smoking prevalence, it would be helpful if the authors included the rate of these subgroups relative to other Asian ethnic groups and gender groups that are fairly close to the targeted population. The authors should also include the prevalence of Vietnamese American LEP.
-It seems that the authors are implying that individuals with LEP are more likely to smoke, however, this was unclear in the introduction. Consider making this connection explicit.
-The argument in lines 44-47 are two separate ideas. Moreover, the proceeding sentence concerning the findings from the systematic review departs from the argument that the paragraph is attempting to make.
Materials and methods:
-In recruiting five community leaders, do these leaders have extensive interaction with Vietnamese Americans who smoke? It’s somewhat of a concern to interview community leaders to provide feedback on smoking cessation if they have no background regarding this topic. It would be helpful for the authors to provide justification regarding the decision to interview these participants.
-Did the authors conduct forward and backward translation of the interviews conducted in the Vietnamese language since all the results were written in English? If so, this was not made clear in the methodology section.
-Did the main coder code the same set of transcripts as the coder who only coded the English transcripts and the Vietnamese transcripts? If so, the authors need to discuss how discrepancies and consensus were reached. If not, provide a reasoning and justification of their approach to the methodology.
-What is the qualitative methodology that the coders used to analyze the qualitative data?
-The authors should also consider including a positionality and reflexivity sub-section of the coders to understand how their own backgrounds may have influenced how the data was analyzed throughout the duration of the study.
Results
-In the limitations section, the authors argued that they reached saturation of the data based on the 16 interviews. Does this mean that for each of the sub-themes and quotes provided that these were consistent findings across most of the interviews or this was not the case? It’s difficult to support that the interviews reached data saturation when there were 4 different subgroups with small sample sizes.
-The findings from the theme “3.4 factors related to all 4 phases of the PBM” did not necessarily reflect all four phases of PBM, some of the findings were still under one of the four phases. The authors should consider eliminating this section.
Discussion
-The logic of the discussion seems somewhat disorganized. I would encourage the authors to choose two to three main findings and providing a thorough discussion. The strength of this study is that it focuses on Vietnamese American perspectives, however, this gets lost in the discussion.
-The authors should lean towards making the explicit connection between their findings and Vietnamese American culture. For example, in the first paragraph of the discussion, the authors discussed “the need to protect their family from the harms of secondhand-smoking…” This goes beyond just self-determination, but other studies on Asian Americans who smoke also tied this to filial piety and familism – that individuals engage in healthy behaviors because it’s a way of paying back to the older generation who sacrificed a lot to move their family to the US. Of course, this is more complicated with the background of Vietnamese refugees. The findings regarding ties to smoking and gender can also be tied back to patriarchy in different Asian cultures. Some literature have made ties to patriarchy as the residual effects of colonialism.
-The primary goal discussed in the introduction section underscored that this study focused on Vietnamese American with ELP, however, how the findings and discussion is tied to populations with ELP who smoke is not discussed thoroughly in the discussion.
Limitations
-The authors should consider discussing further limitations to the study in addition to what was presented. If there was a limitation of “different numbers of participants in each group” that were predominantly male, it is difficult to reach data saturation in qualitative studies.
Round 2
Reviewer 5 Report
The authors have responded adequately to my comments. I congratulate the authors for their manuscript that will be beneficial in expending the literature on smoking in Vietnamese American populations.